# Correlation between Statin Solubility and Mortality in Patients on Chronic Hemodialysis

**DOI:** 10.3390/diagnostics13203290

**Published:** 2023-10-23

**Authors:** Seok-Hui Kang, Gui-Ok Kim, Bo-Yeon Kim, Eun-Jung Son, Jun-Young Do

**Affiliations:** 1Division of Nephrology, Department of Internal Medicine, College of Medicine, Yeungnam University, Daegu 42415, Republic of Korea; kangkang@ynu.ac.kr; 2Quality Assessment Department, Health Insurance Review and Assessment Service, Wonju 26465, Republic of Korea; rrnlfl52@gmail.com (G.-O.K.);; 3Healthcare Review and Assessment Committee, Health Insurance Review and Assessment Service, Wonju 26465, Republic of Korea

**Keywords:** hemodialysis, mortality, statin, solubility

## Abstract

This study aimed to evaluate the effect of statin solubility on the survival of patients undergoing hemodialysis (HD). This retrospective study used laboratory and clinical data from a national HD quality assessment program and claims data (*n* = 53,345). The use of statins was defined as prescription ≥30 days during 6 months of each HD quality assessment period. We divided the patients into three groups based on the use and solubility of statins: No group, patients without a prescription of statins (*n* = 37,944); Hydro group, patients with a prescription of hydrophilic statins (*n* = 2823); and Lipo group, patients with a prescription of lipophilic statins (*n* = 12,578). The 5-year survival rates in the No, Hydro, and Lipo groups were 69.6%, 67.9%, and 67.9%, respectively (*p* < 0.001 for the trend). Multivariable Cox regression analyses showed that the Lipo group had better patient survival than the No group. However, multivariable analyses did not show statistical significance between the Hydro and No or Lipo groups. In all subgroups based on sex, age, presence of diabetes mellitus, and heart disease, the Lipo group had better patient survival than the No group. We identified no significant association between hydrophilic and lipophilic statins and patient survival. However, patients taking lipophilic statins had a modest survival benefit compared with those who did not receive statins.

## 1. Introduction

End-stage renal disease (ESRD) is a critical condition requiring renal replacement therapy (RRT). Hemodialysis (HD), peritoneal dialysis, and renal transplantation are the most important RRT for patients with ESRD. Among the three types of RRT, HD is the most commonly used. Patients undergoing HD have higher mortality and cardiovascular disease rates than the general population [1]. Cardiovascular disease is the leading cause of death in patients undergoing HD [2]. The incidence of cardiovascular disease is a target of current prevention efforts. Traditional cardiovascular risk factors include age, diabetes mellitus (DM), hypertension, or dyslipidemia [3]. However, principles of treatment of traditional risk factors suggested for the general population are not completely applicable to patients on HD. In particular, the benefits and/or risks of statins with regard to lipid-lowering or pleiotropic effects are complex and unresolved. It is unclear whether statin use improves prognosis [4].

Statins are primarily prescribed to reduce low-density lipoprotein cholesterol, by inhibiting HMG-CoA reductase in the liver [5]. Statins are widely used for primary or secondary prevention of cardiovascular diseases and to improve patient survival in various medical conditions. The benefits of statins for cardiovascular diseases or mortality are associated with various pleiotropic effects, such as the stabilization of inflammation or vascular smooth muscle cells, in addition to lipid-lowering effects [6]. Several statins are available, including atorvastatin, simvastatin, rosuvastatin, pitavastatin, pravastatin, lovastatin, and fluvastatin. Statins have different characteristics in terms of potency, dosage, solubility, and drug interactions. Statins are associated with the development of muscle or liver enzyme elevations, or rhabdomyolysis as a side effect, and recent studies have shown a slightly increased risk of new-onset diabetes with the use of statins [5,7]. However, despite concerns regarding certain side effects and long-term use, statins have a relatively well-established safety profile. Previous studies have shown favorable outcomes with the use of statins, which have been one of the most widely prescribed groups of drugs worldwide. Previous studies have provided sufficient evidence regarding the association between statin use and clinical outcomes; however, data regarding the association between these two factors in patients on dialysis are insufficient [4]. Recent randomized studies evaluated the effect of statins on cardiovascular diseases and mortality in patients on dialysis but did not show a definite benefit to the clinical outcomes of statin use [8,9,10]. While the exact cause is not yet clear, it may be attributed to the function of cholesterol as an indicator of malnutrition or inflammation, such as the reverse epidemiology phenomenon [11]. In addition, statin-induced coronary artery calcification is considered a potential cause [12]. A weak association between statin use and clinical outcomes has also been observed in studies that used certain types of statins in patients without dialysis [13,14,15,16]. The relationship between statin use and patient survival and cardiovascular benefits, regardless of HD, has shown inconsistencies, with prior research often attributing these variations to class-wide effects, notably solubility [15,17,18,19,20]. Studies on patients not undergoing HD have evaluated the effect of statin solubility on patient outcomes, but no conclusive results have been obtained. Furthermore, few studies have reported on patients on HD despite the higher prevalence of death and cardiovascular disease compared with patients without HD. This study aimed to evaluate the effect of statin solubility on the survival of patients undergoing HD.

## 2. Materials and Methods

### 2.1. Dataset and Study’s Population

This retrospective study used laboratory and clinical data from a national HD quality assessment program and claims data from the Health Insurance Review and Assessment (HIRA) of Republic of Korea [21,22]. The 4th, 5th, and 6th HD quality assessment programs were conducted from July to December 2013, July to December 2015, and March to August 2018. The programs included patients aged ≥18 years undergoing maintenance HD (≥3 months) and undergoing HD at least twice a week (≥8 per month) (Figure 1). 

We analyzed the HD quality assessment dataset and claims data of all patients on HD. 

The number of patients included in the 4th, 5th, and 6th HD quality assessment programs was 21,846, 35,538, and 31,294, respectively. Among these, we excluded multiple repeat patients or patients with an insufficient dataset (*n* = 32,459), those who underwent HD using a catheter (*n* = 1316), and those who were prescribed two or more statins or who were prescribed statins for <30 days during the 6 months of each assessment (*n* = 1558). Finally, 53,345 patients were included in this study. This study was approved by the Institutional Review Board (IRB) of Yeungnam University Medical Center (approval no. YUMC 2022-01-010). Informed consent was not obtained from the patients because their records and information were anonymized and de-identified before the analysis. 

### 2.2. Study’s Variables

The collected data included age, sex, underlying cause of ESRD, HD vintage (months), and vascular access type. Laboratory data collected during the assessment included hemoglobin (g/dL), Kt/V_urea_, serum albumin (g/dL), serum calcium (mg/dL), serum phosphorus (mg/dL), serum creatinine (mg/dL), predialysis systolic blood pressure (mmHg), predialysis diastolic blood pressure (mmHg), and ultrafiltration volume (L/session). These data were collected monthly, and all laboratory values were averaged from the monthly collected values. Kt/V_urea_ was calculated using the Daugirdas equation, and the formula is as follows: Kt/Vurea = −Ln(R–0.008 × t) + (4–3.5 × R) × UF/W (Ln, natural logarithm; R, postdialysis BUN/predialysis BUN; t, time; UF, ultrafiltration volume; W, postdialysis body weight) [23].

The medication codes are listed in Appendix A. The use of statins was defined as a prescription for ≥30 days during six months of each HD quality assessment period. Solubility was determined as previously described [24]. Briefly, we categorized lipophilic statins as simvastatin, fluvastatin, pitavastatin, lovastatin, and atorvastatin and hydrophilic statins as rosuvastatin or pravastatin. We divided the patients into three groups based on the use and solubility of statins; No group, patients without a prescription of statins; Hydro group, patients with a prescription of hydrophilic statins; and Lipo group, patients with a prescription of lipophilic statins. Additionally, statin dosage was classified into three groups as previously described [25]. Patients who received a mean daily dose of <10 mg/day atorvastatin were defined as having a low statin dosage of statin. The patients who used a mean daily dose of statin ≥40 mg/day of atorvastatin were defined as a high dosage of statin. Patients who received a mean daily dose between the low and high intensities were defined as those who received a moderate dose of statins. 

Medications such as aspirin, renin-angiotensin system blockers, and clopidogrel were also evaluated. If one or more prescriptions were identified a year before the evaluation of the HD quality assessment program, it was defined as the use of medication. These medications are commonly used in patients on HD, especially those using statins, and can be considered confounding factors. Antiplatelet agents, such as aspirin or clopidogrel can be used to treat various cardiovascular diseases, such as myocardial infarction (MI), angina, and peripheral vascular disease. Although these comorbidities were evaluated using ICD-10 codes in our study, diagnoses using ICD-10 codes can be associated with inaccuracies in diagnosis, and the use of these medications for primary prevention may also influence patient outcomes. Additionally, the use of renin-angiotensin system blockers is associated with favorable outcomes in patients taking the drug for secondary prevention of cardiovascular diseases, and the use of renin-angiotensin system blockers for primary prevention can also be associated with favorable outcomes despite incomplete conclusions regarding the association between the use of renin-angiotensin system blockers and patient mortality. Therefore, we used these factors as covariates to reduce their confounding effects and limitations associated with the inaccuracy of diagnosis.

The presence of comorbidities was evaluated for a year before the HD quality assessment program. Comorbidities were defined using the codes described by Quan et al. [26,27]. The Charlson Comorbidity Index (CCI) included 17 comorbidities. In our study, the CCI included 17 comorbidities; MI, congestive heart failure (CHF), peripheral vascular disease, cerebrovascular disease, dementia, chronic pulmonary disease, rheumatologic disease, peptic ulcer disease, mild liver disease, DM without chronic complications, hemiplegia, renal disease, DM with chronic complications, any malignancy, moderate-to-severe liver disease, metastatic tumor, and acquired immune deficiency syndrome. The presence of comorbidities was defined using the ICD-10 codes listed in Appendix A. All patients were considered to have renal disease due to HD (two points for moderate or severe renal disease). Finally, the CCI score was calculated using the 17 comorbidities mentioned above. 

Outcomes were monitored until April 2022. If the patient was transferred for peritoneal dialysis or kidney transplantation, the date was the endpoint of follow-up, and the data were censored. Clinical outcomes, except death, were defined during the follow-up using electronic data. The codes for censoring were O7072, O7071, and O7061 for peritoneal dialysis, and R3280 for kidney transplantation. Patient-death data were obtained from the HIRA database. 

### 2.3. Statistical Analyses

Data were analyzed using the SAS Enterprise Guide version 7.1 (SAS Institute, Cary, NC, USA) or R software (version 3.5.1; R Foundation for Statistical Computing, Vienna, Austria). Categorical variables are presented as numbers and percentages, whereas continuous variables are presented as the means ± standard deviations. Pearson’s χ^2^ test or Fisher’s exact test was used to analyze categorical variables. For continuous variables, means were compared using a one-way analysis of variance, followed by Tukey’s post hoc test. Survival estimates were calculated using Kaplan–Meier curves and Cox regression analysis. *p*-Values for comparison of survival curves were determined using the log-rank test. Multivariable Cox regression analyses were adjusted for age, sex, type of vascular access, underlying cause of ESRD, CCI score, HD vintage, ultrafiltration volume, Kt/V_urea_, hemoglobin, serum albumin, serum creatinine, serum phosphorus, serum calcium, systolic blood pressure, diastolic blood pressure, use of aspirin, renin-angiotensin system blockers, clopidogrel, and statin dosage. All the variables associated with patient survival were selected as covariates. All baseline characteristics were well-known factors associated with patient survival and were selected as covariates. Multivariable Cox regression analyses were performed using the enter mode. Statistical significance was set at *p* < 0.05.

## 3. Results

### 3.1. Participant’s Clinical Characteristics

The number of patients in the No, Hydro, and Lipo groups was 37,944, 2823, and 12,578, respectively (Table 1). 

The patients in the No group had a higher proportion of male sex and arteriovenous fistula; a lower proportion of DM, MI, or CHF; and the use of renin-angiotensin system blockers, aspirin, or clopidogrel than those in the other two groups. The patients in the No group had lower CCI scores and hemoglobin levels and were younger than those in the other two groups. Additionally, patients in the No group had higher HD vintages, follow-up durations, phosphorus levels, diastolic blood pressure, and serum creatinine levels than those in the other two groups. The Hydro group received a higher proportion of high-dose statins.

### 3.2. Survival Analyses

The number of patients in the survivor, death, PD, or kidney transplantation subgroups at the end-point of follow-up was as follows: 19,995 (52.7%), 14,726 (38.8%), 143 (0.4%), and 3080 (8.1%), respectively, in the No group; 1590 (56.3%), 1044 (37.0%), 10 (0.4%), and 179 (6.3%), respectively, in the Hydro group; and 6797 (54.0%), 4957 (39.4%), 33 (0.3%), and 791 (6.3%), respectively; in the Lipo group, respectively (*p* < 0.001).

The 5-year survival rates in the No, Hydro, and Lipo groups were 69.6%, 67.9%, and 67.9%, respectively (Figure 2, *p* < 0.001 for the trend). 

The No group had better patient survival rates than those of the other two groups, and no survival difference was observed between the Hydro and Lipo groups. Univariate Cox regression analyses showed that the hazard ratios were 1.10 (95% confidence interval [CI], 1.03–1.17, *p* = 0.005) in the Hydro group and 1.09 (95% CI, 1.06–1.13, *p* < 0.001) in the Lipo group compared with the No group (Table 2). 

Multivariable Cox regression analyses showed that the Lipo group had better patient survival than the No group. However, multivariable analyses did not show statistical significance between the Hydro and No or Lipo groups.

Subgroup analyses were performed based on sex, age, the presence of DM, heart disease (MI or CHF), and statin dosage. In all subgroups based on sex, age, the presence of DM, and heart disease, the Lipo group exhibited better patient survival than the No group (Appendix A). There were no significant differences in patient survival between the Hydro and Lipo groups in the three dosage subgroups.

## 4. Discussion

We analyzed 53,345 patients who underwent HD quality assessment in the Republic of Korea. Our results revealed no significant differences in all-cause mortality between hydrophilic and lipophilic statins. However, lipophilic statins were associated with modest survival benefits compared with those without statins. These trends were similar in the subgroup analyses.

With the emergence of studies examining the association between cholesterol and cardiovascular diseases as well as mortality, efforts have been made to efficiently reduce cholesterol levels, especially low-density lipoprotein cholesterol. The discovery of HMG-CoA reductase, a rate-controlling enzyme in cholesterol synthesis, has paved the way for these efforts. Researchers in Japan, including Akira Endo, successfully isolated compactin, a competitive inhibitor of HMG-CoA reductase, from fungi, and its effectiveness was confirmed when it was first applied in humans [28]. Subsequently, lovastatin became the first statin to be marketed, and over time, a total of six statins, including two semi-synthetic statins (simvastatin and pravastatin) and four synthetic statins (fluvastatin, rosuvastatin, pitavastatin, and atorvastatin), were developed and are currently in use [29]. Previous guidelines have emphasized that hypercholesterolemia is a well-known cardiovascular risk factor, and the use of statins for primary prevention is recommended based on atherosclerotic cardiovascular disease risk assessment [30]. Furthermore, two guidelines from the United States and Europe noted a direct correlation between cholesterol levels and atherosclerotic cardiovascular diseases in the general population, recommending statin therapy for secondary prevention [31]. Statins are commonly used as first-line therapies for lipid disorders, particularly in patients with elevated low-density lipoprotein cholesterol.

Statins can be divided into two classes based on solubility. Lipophilic statins exhibit high cell membrane permeability. However, experimental and clinical studies have revealed contradictory results regarding the association between statin solubility and clinical effects. Previous studies have shown that lipophilic statins have better cell membrane penetration into extrahepatic cells than into hepatocytes, which may be associated with better pleiotropic and/or cholesterol-independent effects compared with those of hydrophilic statins [32,33,34]. Therefore, lipophilic statins have been associated with improved clinical outcomes. However, other studies have shown that lipophilic statins exhibit greater penetration into myocytes, inhibiting isoprenoid and coenzyme Q10 synthesis [19,35]. These drugs can decrease myocyte contractility, and lipophilic statins are associated with poor clinical outcomes.

The results of the association between the solubility of statins and clinical outcomes are inconsistent. Bielecka-Dabrowa et al. analyzed 17 studies, including two randomized controlled trials, in patients with heart failure [17]. The authors reported better outcomes in patients treated with lipophilic statins than in those treated with hydrophilic statins. A meta-analysis of coronary artery disease included 11 randomized studies and revealed similar outcomes between hydrophilic and lipophilic statins in patients with this disease [18]. Wang et al. used insurance data from Taiwan to include patients undergoing chronic dialysis [19]. They demonstrated that hydrophilic statins were associated with a lower cardiovascular event rate than lipophilic statins; however, no statistical significance in all-cause mortality was observed between the two groups. Kim et al. included acute MI and chronic kidney disease with an estimated glomerular filtration rate of <60 mL/min/1.73 m^2^ [20]. They also reported lower rates of major cardiovascular events and all-cause mortality in patients treated with hydrophilic statins than in those treated with lipophilic statins.

Our study revealed no significant difference in patient survival between the Hydro group and the No or Lipo groups. However, the Lipo group had modest benefits in patient survival compared with the No group. These results are consistent with those presented by Bielecka-Dabrowa et al., who enrolled patients with heart failure regardless of renal function. A previous study using patients on chronic dialysis from a Taiwanese population showed significantly better all-cause mortality in statin users than in non-statin users and lower cardiovascular events in hydrophilic statin use than in lipophilic statin use [19]. However, all-cause mortality did not differ between the two groups. Our study and that by Wang et al. originated in an Asian population. However, the study by Wang et al. had some limitations. First, they did not include the clinical or laboratory data commonly observed in patients undergoing dialysis. Consequently, they did not fully adjust for the confounding factors associated with patient outcomes. Second, the authors did not differentiate between PD and HD. Third, the definitions of statin use were relatively limited. Statin use was defined as statin use for at least 28 days during the follow-up period. Our study has some strengths compared with Wang’s study. Our study included only patients on maintenance HD and various clinical and laboratory data associated with patient outcomes. Therefore, multivariable analysis could be performed using additional confounding factors.

In our study, there were differences in the baseline characteristics among the three groups. The No group exhibited fewer comorbidities and a tendency toward a younger age. Intergroup heterogeneity is an inherent limitation, particularly in retrospective observational studies, such as ours. Intergroup heterogeneity can be a significant confounding factor when analyzing intergroup differences in outcomes. To mitigate this, we attempted various approaches, such as multivariable and subgroup analyses, which yielded similar results. However, it remains challenging to completely eliminate the effects of intergroup heterogeneity and establish a clear cause–effect relationship. Notably, the No group exhibited fewer comorbidities and a tendency toward a younger age profile, both of which are factors associated with better outcomes. Despite the presence of these favorable prognostic factors in the No group, it is noteworthy that survival rates were lower in the No group than in the Lipo group in both univariate and multivariable analyses. These findings suggest that the benefits of lipophilic statins on survival should be interpreted in a more meaningful manner. Ultimately, randomized prospective studies are required to effectively address this heterogeneity.

Our study utilized laboratory test results for HD quality assessment and associated claims data. Although we were able to confirm the medication prescription, we did not access data on the history of statin initiation. The HD quality assessment program primarily focused on clearly proven risk factors for mortality, such as anemia, calcium/phosphate levels, blood pressure, and adequacy of dialysis. Lipid profile was not included in this program because of the lack of a clear association between mortality and dyslipidemia. In addition, assessing the indications for statin use is crucial for evaluating statin use and patient outcomes. The Kidney Disease: Improving Global Outcomes (KDIGO) guidelines recommend that in patients already receiving statins at the time of dialysis initiation, these agents can be continued, and in patients on HD, these agents cannot be initiated [4]. However, this recommendation was limited to the use of statins for primary prevention. Although the KDIGO guidelines do not provide recommendations for secondary prevention, recent research has suggested that using statins for secondary prevention after cardiovascular events may yield favorable outcomes [36,37,38]. Although we did not include data on the history of statin initiation, considering the limited efficacy of statins for primary prevention and the potentially favorable results for secondary prevention, we believe that many patients in our study may have used statins for secondary prevention. However, further analyses using data sources that provide information on medication use and indications, such as multicenter or registry studies, were warranted to confirm the association between statin use for secondary prevention and outcomes. 

However, our study had certain limitations. First, our study was retrospective, and there were large differences in the sample size and baseline characteristics among the three groups. In particular, the sample size was small in the Hydro group, and the No group differed in baseline characteristics compared with those of the other two groups. Second, comorbidities, including heart disease, and statin use were evaluated using the claims data. Our study included only information about prescription status, rather than actual medication intake, to distinguish patient groups. Therefore, patients were categorized based on whether they had been prescribed medications and the number of days for which these prescriptions were issued. While an analysis based on actual medication intake could provide greater accuracy, our study exclusively utilized claims data related to prescription status, prescription quantities, prescription durations, and associated claims data, without including data on patients’ actual consumption or chart reviews. This approach could introduce a potential discrepancy between prescribed medications and their actual consumption, which is considered a significant limitation. Despite these limitations, it is worth noting that recent population-based studies have increasingly utilized claims data to classify and analyze patients based on their prescriptions. These studies serve as preliminary investigations and are conducted before randomized trials or similar studies are performed. In addition, our study did not include laboratory data on lipid status and etiology of statin use. The effects of statins on patient survival may differ between patients taking medications for MI or CHF and those taking medications for simple dyslipidemia. Third, our study did not include data on the cause of death or detailed data on heart function, such as heart rate, left ventricular hypertrophy, cardiac mass, or ejection fraction. The benefits of statins are mainly related to cardiovascular disease, and information on cardiovascular death and/or heart function would be useful for identifying differences in statin solubility beyond all-cause mortality. 

In conclusion, we identified no significant association between hydrophilic and lipophilic statins and patient survival. However, patients taking lipophilic statins had a modest survival benefit compared with those who did not receive statins. Thus, if statin is indicated, the use of lipophilic statins may be indicated in patients on HD, despite the non-superiority between hydrophilic and lipophilic statins. Considering the limitations of this study, the results should be interpreted with caution. Further randomized prospective studies are required to determine whether statin solubility is associated with outcomes in patients undergoing HD. 

## Figures and Tables

**Figure 1 diagnostics-13-03290-f001:**
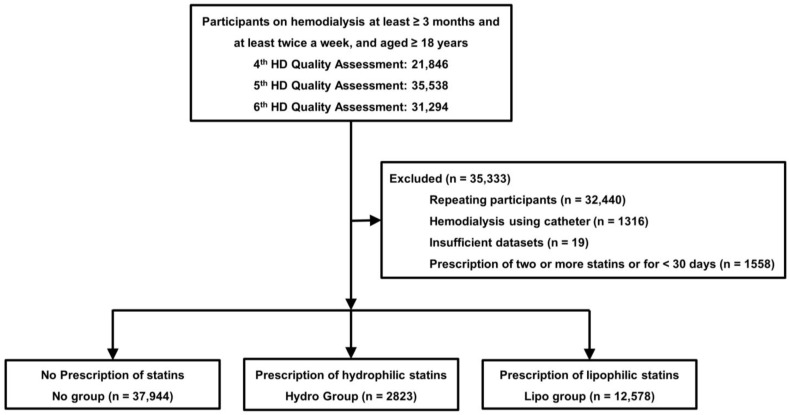
Study flowchart. Abbreviations: HD, hemodialysis.

**Figure 2 diagnostics-13-03290-f002:**
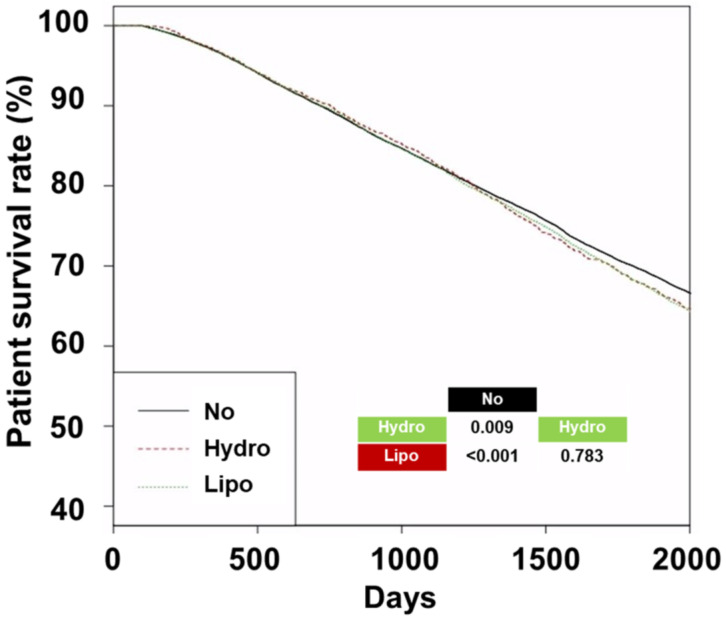
Kaplan–Meier curves of patient survival for groups. The *p*-values for pairwise comparison or trend with log-rank tests were added to the lower right corner of the graph. Abbreviations: No, patients without a prescription of statins; Hydro, patients with a prescription of hydrophilic statins; Lipo, patients with a prescription of lipophilic statins.

**Table 1 diagnostics-13-03290-t001:** Patient clinical characteristics.

	No Group (*n* = 37,944)	Hydro Group (*n* = 2823)	Lipo Group (*n* = 12,578)	*p*
Age (years)	59.4 ± 13.3	62.1 ± 11.9 *	62.1 ± 12.1 *	<0.001
Sex (male, %)	23,450 (61.8%)	1555 (55.1%)	6906 (54.9%)	<0.001
Hemodialysis vintage (months)	56.4 ± 59.0	40.4 ± 46.2 *	42.2 ± 47.4 *	<0.001
Underlying causes of ESRD				<0.001
Diabetes mellitus	14,642 (38.6%)	1589 (56.3%)	7028 (55.9%)	
Hypertension	10,707 (28.2%)	626 (22.2%)	2726 (21.7%)	
Glomerulonephritis	4453 (11.7%)	190 (6.7%)	1048 (8.3%)	
Others	3538 (9.3%)	182 (6.4%)	785 (6.2%)	
Unknown	4604 (12.1%)	236 (8.4%)	991 (7.9%)	
CCI score	7.2 ± 2.9	8.4 ± 2.7 *	8.1 ± 2.8 *^#^	<0.001
Follow-up duration (months)	62.2 ± 29.2	56.1 ± 25.1 *	59.0 ± 27.1 *^#^	<0.001
Type of vascular access				<0.001
Arteriovenous fistula	32,538 (85.8%)	2366 (83.8%)	10,596 (84.2%)	
Arteriovenous graft	5406 (14.2%)	457 (16.2%)	1982 (15.8%)	
Kt/V_urea_	1.53 ± 0.27	1.54 ± 0.27	1.53 ± 0.27 *	0.006
Ultrafiltration volume (L/session)	2.29 ± 0.96	2.22 ± 0.91 *	2.23 ± 0.94 *	<0.001
Hemoglobin (g/dL)	10.6 ± 0.8	10.7 ± 0.7 *	10.7 ± 0.7 *	<0.001
Serum albumin (g/dL)	3.99 ± 0.34	3.99 ± 0.34	3.98 ± 0.33 *	0.041
Serum phosphorus (mg/dL)	5.0 ± 1.4	4.8 ± 1.3 *	4.8 ± 1.3 *	<0.001
Serum calcium (mg/dL)	8.9 ± 0.8	8.8 ± 0.7 *	8.9 ± 0.8 *	<0.001
Systolic blood pressure (mmHg)	141 ± 16	142 ± 16	141 ± 16	0.580
Diastolic blood pressure (mmHg)	79 ± 9	76 ± 10 *	77 ± 10 *^#^	<0.001
Serum creatinine (mg/dL)	9.67 ± 2.77	8.94 ± 2.63 *	9.09 ± 2.60 *^#^	<0.001
Use of RASB	11,161 (29.4%)	876 (31.0%)	4228 (33.6%)	<0.001
Use of aspirin	13,898 (36.6%)	1591 (56.4%)	7146 (56.8%)	<0.001
Use of clopidogrel	4268 (11.2%)	837 (29.6%)	3484 (27.7%)	<0.001
MI or CHF	15,945 (42.0%)	1606 (56.9%)	6524 (51.9%)	<0.001
Dosage of statin				<0.001
Low	–	298 (10.6%)	402 (3.2%)	
Moderate	–	2287 (81.0%)	11,873 (94.4%)	
High	–	238 (8.4%)	303 (2.4%)	

Data are expressed as mean ± standard deviation for continuous variables and as numbers (percentages) for categorical variables. *p*-Values were tested using a one-way analysis of variance, followed by Tukey post hoc test, and Pearson’s χ^2^ test for categorical variables. Abbreviations: CCI, Charlson Comorbidity Index; CHF, congestive heart failure; ESRD, end-stage renal disease; MI, myocardial infarction; RASB, renin-angiotensin system blockers. * *p* < 0.05 vs. No statin, ^#^
*p* < 0.05 vs. Hydrophilic.

**Table 2 diagnostics-13-03290-t002:** Cox regression analyses for patient survival in hemodialysis patients.

	Univariate		Multivariable	
	HR (95% CI)	*p*	HR (95% CI)	*p*
Group				
Ref: No group				
Hydro group	1.10 (1.03–1.17)	0.005	0.95 (0.88–1.03)	0.215
Lipo group	1.09 (1.06–1.13)	<0.001	0.92 (0.88–0.96)	<0.001
Ref: Hydro group				
Lipo group	0.99 (0.93–1.06)	0.783	0.97 (0.89–1.06)	0.511
Age (increase per 1 year)	1.06 (1.06–1.06)	<0.001	1.06 (1.06–1.06)	<0.001
Sex (ref: male)	0.87 (0.84–0.89)	<0.001	0.75 (0.72–0.78)	<0.001
Underlying cause of ESRD (ref: DM)	0.81 (0.80–0.82)	<0.001	0.90 (0.89–0.91)	<0.001
Vascular access (ref: arteriovenous fistula)	1.51 (1.46–1.56)	<0.001	1.18 (1.13–1.23)	<0.001
Hemodialysis vintage (increase per 1 month)	0.99 (0.99–1.00)	0.100	1.00 (1.00–1.01)	<0.001
CCI score (increase per 1 score)	1.14 (1.13–1.14)	<0.001	1.06 (1.06–1.07)	<0.001
UFV (increase per 1 kg/session)	0.92 (0.90–0.93)	<0.001	1.07 (1.05–1.09)	<0.001
KtV_urea_ (increase per 1 unit)	0.91 (0.86–0.96)	<0.001	0.80 (0.75–0.87)	<0.001
Hemoglobin (increase per 1 g/dL)	0.87 (0.85–0.88)	<0.001	0.91 (0.89–0.93)	<0.001
Serum albumin (increase per 1 g/dL)	0.37 (0.36–0.39)	<0.001	0.63 (0.59–0.66)	<0.001
Serum creatinine (increase per 1 mg/dL)	0.87 (0.86–0.87)	<0.001	0.94 (0.93–0.94)	<0.001
Serum phosphorus (increase per 1 mg/dL)	0.85 (0.84–0.86)	<0.001	1.04 (1.03–1.06)	<0.001
Serum calcium (increase per 1 mg/dL)	0.93 (0.92–0.95)	<0.001	1.06 (1.04–1.08)	<0.001
SBP (increase per 1 mmHg)	1.01 (1.01–1.01)	<0.001	1.01 (1.00–1.01)	<0.001
DBP (increase per 1 mmHg)	0.98 (0.98–0.98)	<0.001	1.00 (1.00–1.01)	0.020
Use of renin angiotensin system blocker	1.15 (1.12–1.18)	<0.001	1.01 (0.98–1.05)	0.584
Use of clopidogrel	1.53 (1.49–1.59)	<0.001	1.15 (1.10–1.20)	<0.001
Use of aspirin	1.16 (1.13–1.19)	<0.001	0.96 (0.93–0.99)	0.016
MI or CHF	1.49 (1.45–1.53)	<0.001	1.05 (1.01–1.09)	0.011
Dosage of statin (ref: low)				
Moderate	0.95 (0.85–1.07)	0.404	0.97 (0.84–1.12)	0.678
High	1.11 (0.93–1.32)	0.254	1.05 (0.84–1.30)	0.679

Multivariable analysis was adjusted for age, sex, underlying cause of ESRD, vascular access, hemodialysis vintage, CCI score, ultrafiltration volume, Kt/V_urea_, hemoglobin, serum albumin, serum creatinine, serum phosphorus, serum calcium, systolic blood pressure, diastolic blood pressure, use of renin-angiotensin system blockers, clopidogrel, aspirin, MI or CHF, dosage of statin, and was performed using enter mode. Abbreviations: CCI, Charlson Comorbidity Index; CHF, congestive heart failure; ESRD, end-stage renal disease; CI, confidence interval; DBP, diastolic blood pressure; DM, diabetes mellitus; HR, hazard ratio; MI, myocardial infarction; SBP, systolic blood pressure; UFV, ultrafiltration volume.

## Data Availability

Raw data were generated by the Health Insurance Review and Assessment Service. The database can be requested from the Health Insurance Review and Assessment Service by sending a study proposal, including the purpose of the study, study design, and duration of analysis through on the website (https://www.hira.or.kr, accessed on 11 September 2023). The authors cannot distribute the data without permission.

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
