# Peer review of "Correlation between Statin Solubility and Mortality in Patients on Chronic Hemodialysis"

_diagnostics, 2023, doi:10.3390/diagnostics13203290_

Round 1
Reviewer 1 Report
Dear Authors, I have read with interest your manuscript. The paper addresses an very interesting issue regarding statin therapy in dialyzed patients considering that there are no very clear recommendations in this regard and the effects of these drugs on dialyzed patients are not very well studied.
I would like to address a few suggestions/ questions:
I suggest you, starting from the idea of ​​this study, to design a prospective study, in a more heterogenous population; in your group description, the population in the No group,is younger, with fewer comorbidities, as compared with the other groups, which have more comorbidities and are less younger. The conclusion is that the three groups are not heterogenous, hence the very high risk of confusion.
Also, there are no data on the lab values of lipids in your patients and on the history of initiation of statins in the patients in the two groups who have this treatment and I think this is very important, because the recommendation is to continue statin therapy in a patients with CKD with history of stroke or infarction or vascular disease.
This topic is very interesting to discus becausewe neeed to use all the available resource to descrease the cardiovascular risk of dialyzed patients.
Author Response
Dear Authors, I have read with interest your manuscript. The paper addresses an very interesting issue regarding statin therapy in dialyzed patients considering that there are no very clear recommendations in this regard and the effects of these drugs on dialyzed patients are not very well studied.
I would like to address a few suggestions/ questions:
I suggest you, starting from the idea of ​​this study, to design a prospective study, in a more heterogenous population; in your group description, the population in the No group,is younger, with fewer comorbidities, as compared with the other groups, which have more comorbidities and are less younger. The conclusion is that the three groups are not heterogenous, hence the very high risk of confusion.
Answer: Thank you for your comments. In our study, there were differences in the baseline characteristics among the three groups. The No group exhibited fewer comorbidities and a tendency toward a younger age. Intergroup heterogeneity is an inherent limitation, particularly in retrospective observational studies, such as ours. Intergroup heterogeneity can be a significant confounding factor when analyzing intergroup differences in outcomes. To mitigate this, we attempted various approaches, such as multivariate and subgroup analyses, which yielded similar results. However, it remains challenging to completely eliminate the effects of intergroup heterogeneity and establish a clear cause-effect relationship. Notably, the No group exhibited fewer comorbidities and a tendency toward a younger age profile, both of which are factors associated with better outcomes. Despite the presence of these favorable prognostic factors in the No group, it is noteworthy that survival rates were lower in the No group than in the Lipo group in both univariate and multivariate analyses. These findings suggest that the benefits of lipophilic statins on survival should be interpreted in a more meaningful manner. Ultimately, randomized prospective studies are require to effectively address this heterogeneity.
We have added these comments in the Discussion section.
Also, there are no data on the lab values of lipids in your patients and on the history of initiation of statins in the patients in the two groups who have this treatment and I think this is very important, because the recommendation is to continue statin therapy in a patients with CKD with history of stroke or infarction or vascular disease.
Answer: Thank you for your comments. Our study utilized laboratory test results for HD quality assessment and associated claims data. Although we were able to confirm the medication prescriptions, we did not access data on the history of statin initiation. The HD quality assessment program primarily focused on clearly proven risk factors for mortality, such as anemia, calcium/phosphate levels, blood pressure, and adequacy of dialysis. Lipid profile was not included in this program because of the lack of a clear association between mortality and dyslipidemia. In addition, assessing the indications for statin use is crucial for evaluating statin use and patient outcomes. The KDIGO guidelines recommend that in patients already receiving statins at the time of dialysis initiation, these agents can be continued, and in patients on HD, these agents cannot be initiated [1]. However, this recommendation was limited to the use of statins for primary prevention. Although the KDIGO guidelines do not provide recommendations for secondary prevention, recent research has suggested that using statins for secondary prevention after cardiovascular events may yield favorable outcomes [2-4]. Although we did not include data on the history of statin initiation, considering the limited efficacy of statins for primary prevention and the potentially favorable results for secondary prevention, we believe that many patients in our study may have used statins for secondary prevention. However, further analyses using data sources that provide information on medication use and indications, such as multicenter or registry studies, are warranted to confirm the association between statin use for secondary prevention and outcomes.
We have added these comments in the Discussion section.
Added references
[1] Kidney Disease: Improving Global Outcomes (KDIGO) Lipid Work Group. KDIGO clinical practice guideline for lipid management in chronic kidney disease. Kidney Int Suppl. 2013, 3, 259-305.
[2] Chung, C.M.; Lin, M.S.; Chang, C.H.; Cheng, H.W.; Chang, S.T.; Wang, P.C.; Chang, H.Y.; Lin, Y.S. Moderate to high intensity statin in dialysis patients after acute myocardial infarction: A national cohort study in Asia. Atherosclerosis. 2017, 267, 158-166.
[3] Lin, T.Y.; Hsieh, T.H.; Hung, S.C. Association of secondary prevention medication use after myocardial infarction with mortality in hemodialysis patients. Clinical kidney journal. 2022, 15, 2135-2143.
[4] Lee, M.; Choi, W.J.; Lee, Y.; Lee, K.; Park, M.W.; Myong, J.P.; Kim, D.W. Association between statin therapy and mortality in patients on dialysis after atherosclerotic cardiovascular diseases. Scientific reports. 2023, 13, 10940.
This topic is very interesting to discuss because we need to use all the available resource to decrease the cardiovascular risk of dialyzed patients.

Author Response
The aim of the study titled ’Correlation between the statin solubility and mortality in patients on chronic hemodialysis’ is notable. Cardiovascular complications are the main cause of morbidity and mortality in CKD patients, and that is why statins are used very often in this group. The manuscript is well written, however, I have some comments.
Introduction
- Line 33: Patients with HD patients have higher mortality and cardiovascular disease rates than those of the general population. Please, improve English and style in this sentence, e.g. Patients treated with HD have higher morbidity and mortality due to cardiovascular complications.
Answer: Thank you for your comment. We have revised the relevant sentence to “Patients undergoing HD have higher mortality and cardiovascular disease rates than the general population.”
- Line 34: . Cardiovascular diseases are the most common cause of death in patients on HD. This sentence is the same as the previous one.
Answer: Thank you for your comment. We have revised the relevant sentence to “Cardiovascular disease is the leading cause of death in patients undergoing HD.”
- Please, improve your style, line 44 and 46: previous studies is repeated. Please, change it.
Answer: Thank you for your comment. We have revised two sentences to one as follows: The relationship between statin use and patient survival and cardiovascular benefits, regardless of HD, has shown inconsistencies, with prior research often attributing these variations to class-wide effects notably solubility [5-8].
- Line 47: s. Furthermore, few studies have reported on patients on HD despite the higher prevalence of death and cardiovascular disease compared to studies on patients without HD. Please, add references.
Answer: Thank you for your comments. We have added a reference.
- The introduction is too short. You should write more information on cardiovascular complications in CKD patients and then the second paragraph should be enriched in more information about statins, e.g. the groups of statins, their side effects, what is the safety of the use of statins in the general population and in CKD patients. Although the aim of the study is very interesting, the introduction should be improved and written better to encourage readers to follow the text. Please, change the introduction section.
Answer: Thank you for your comments. We have revised the second paragraph of Introduction section as follows:
Statins are primarily prescribed to reduce low-density lipoprotein cholesterol, by inhibiting HMG-CoA reductase in the liver [1]. Statins are widely used for primary or secondary prevention of cardiovascular diseases and to improve patient survival in various medical conditions. The benefits of statins for cardiovascular diseases or mortality are associated with various pleiotropic effects, such as the stabilization of inflammation or vascular smooth muscle cells, in addition to lipid-lowering effects [2]. Several statins are available, including atorvastatin, simvastatin, rosuvastatin, pitavastatin, pravastatin, lovastatin, and fluvastatin. Statins have different characteristics in terms of potency, dosage, solubility, and drug interactions. Statins are associated with the development of muscle or liver enzyme elevations, or rhabdomyolysis as side effects, and recent studies have shown a slightly increased risk of new-onset diabetes with the use of statins [1,3]. However, despite concerns regarding certain side effects and long-term use, statins have a relatively well-established safety profile. Previous studies have shown favorable outcomes with the use of statins, which have been one of the most widely prescribed groups of drugs worldwide. Previous studies have provided sufficient evidence regarding the association between statin use and clinical outcomes; however, data regarding the association between these two factors in patients on dialysis are insufficient [4]. Recent randomized studies evaluated the effect of statins on cardiovascular diseases and mortality in patients on dialysis but did not show a definite benefit on the clinical outcomes of statin use [5-7]. While the exact cause is not yet clear, it may be attributed to the function of cholesterol as an indicator of malnutrition or inflammation, such as the reverse epidemiology phenomenon [8]. In addition, statin-induced coronary artery calcification is considered a potential cause [9]. A weak association between statin use and clinical outcomes has also been observed in studies that used certain types of statins in patients without dialysis [10-13]. The relationship between statin use and patient survival and cardiovascular benefits, regardless of HD, has shown inconsistencies, with prior research often attributing these variations to class-wide effects, notably solubility [12,14-17]. Studies on patients not undergoing HD have evaluated the effect of statin solubility on patient outcomes, but no conclusive results have been obtained. Furthermore, few studies have reported on patients on HD despite the higher prevalence of death and cardiovascular disease compared to patients without HD. This study aimed to evaluate the effect of statin solubility on the survival of patients undergoing HD.
Added references
[1] Ferri, N.; Corsini, A. Clinical Pharmacology of Statins: an Update. Curr Atheroscler Rep. 2020, 22(7), 26.
[2] Oesterle, A.; Laufs, U.; Liao, J.K. Pleiotropic Effects of Statins on the Cardiovascular System. Cir Res. 2017, 120(1), 229-243.
[3] Mansi, I.A.; Sumithran, P.; Kinaan, M. Risk of diabetes with statins. BMJ. 2023, 381, e071727.
[4] Kidney Disease: Improving Global Outcomes (KDIGO) Lipid Work Group. KDIGO clinical practice guideline for lipid management in chronic kidney disease. Kidney Int Suppl. 2013, 3, 259-305.
[5] Wanner, C.; Krane, V.; Marz, W.; Olschewski, M.; Mann, J.F.; Ruf, G.; Ritz, E.; German Diabetes and Dialysis Study Investigators. Atorvastatin in patients with type 2 diabetes mellitus undergoing hemodialysis. N Engl J Med. 2005, 353, 238-248.
[6] Fellstrom BC, Jardine AG, Schmieder RE, Holdaas H, Bannister K, Beutler J, et al. Rosuvastatin and cardiovascular events in patients undergoing hemodialysis. N Engl J Med. 2009;360:1395-1407.
[7] Baigent, C.; Landray, M.J.; Reith, C.; Emberson, J.; Wheeler, D.C.; Tomson, C., et al. The effects of lowering LDL cholesterol with simvastatin plus ezetimibe in patients with chronic kidney disease (Study of Heart and Renal Protection): a randomised placebo-controlled trial. Lancet. 2011, 377, 2181-2192.
[8] Kalantar-Zadeh, K.; Block, G.; Humphreys, M.H.; Kopple, J.D. Reverse epidemiology of cardiovascular risk factors in maintenance dialysis patients. Kidney Int. 2003, 63, 793-808.
[9] Chen, Z.; Qureshi, A.R.; Parini, P.; Hurt-Camejo, E.; Ripsweden, J.; Brismar, T.B.; Barany, P.; Jaminon, A.M.; Schurgers, L.J.; Heimbürger, O.; Lindholm, B.; Stenvinkel, P. Does statins promote vascular calcification in chronic kidney disease? Eur J Clin Invest. 2017, 47, 137-148.
[10] Arnaboldi, L.; Corsini, A. Do structural differences in statins correlate with clinical efficacy?. Curr Opin Lipidol. 2010, 21, 298-304.
[11] Backes, J.M.; Howard, P.A.; Ruisinger, J.F.; Moriarty, P.M. Does simvastatin cause more myotoxicity compared with other statins?. Ann Pharmacother. 2009, 43, 2012-2020.
[12] Kim, M.C.; Ahn, Y.; Jang, S.Y.; Cho, K.H.; Hwang, S.H.; Lee, M.G.; Ko, J.S.; Park, K.H.; Sim, D.S.; Yoon, N.S.; Yoon, H.J.; Kim, K.H.; Hong, Y.J.; Park, H.W.; Kim, J.H.; Jeong, M.H.; Cho, J.G.; Park, J.C.; Kang, J.C. Comparison of clinical outcomes of hydrophilic and lipophilic statins in patients with acute myocardial infarction. Korean J Intern Med. 2011, 26, 294-303.
[13] Maruyama, T.; Takada, M.; Nishibori, Y.; Fujita, K.; Miki, K.; Masuda, S.; Horimatsu, T.; Hasuike, T. Comparison of preventive effect on cardiovascular events with different statins. -The CIRCLE study-. Circ J. 2011, 75, 1951-1959.
[14] Bielecka-Dabrowa, A.; Bytyçi, I.; Von Haehling, S.; Anker, S.; Jozwiak, J.; Rysz, J.; Hernandez, A.V.; Bajraktari, G.; Mikhailidis, D.P.; Banach, M. Association of statin use and clinical outcomes in heart failure patients: a systematic review and meta-analysis. Lipids Health Dis. 2019, 18, 188.
[15] Bytyçi, I.; Bajraktari, G.; Bhatt, D.L.; Morgan, C.J.; Ahmed, A.; Aronow, W.S.; Banach, M.; Lipid and Blood Pressure Meta-analysis Collaboration (LBPMC) Group. Hydrophilic vs lipophilic statins in coronary artery disease: A meta-analysis of randomized controlled trials. J Clin Lipidol. 2017, 11, 624-637.
[16] Wang. S.W.; Li, L.C.; Su, C.H.; Yang, Y.H.; Hsu, T.W.; Hsu, C.N. Association of Statin and Its Lipophilicity With Cardiovascular Events in Patients Receiving Chronic Dialysis. Clin Pharmacol Ther. 2020, 107, 1312-1324.
[17] Kang, M.H.; Kim, W.; Kim, J.S.; Jeong, K.H.; Jeong, M.H.; Hwang, J.Y.; Hur, S.H.; Hwang, H.S. Hydrophilic Versus Lipophilic Statin Treatments in Patients With Renal Impairment After Acute Myocardial Infarction. J Am Heart Assoc. 2022, 11, e024649.
Materials and Methods
- Please, answer the question why the informed consent was not taken for the study. In all studies, the results are anonymous, but if the researchers want to use the laboratory data of patients the informed consent should be written. So why was the study approved by Institutional Review Board (IRB) of Yeungnam? I do not understand, please explain.
Answer: Thank you for your comments.
Data of HD quality assessment program were collected in an anonymous manner. Our study was a retrospective analysis of data collected anonymously for public purposes from an HD quality assessment program. The IRB of Yeungnam University Medical Center approved the study protocol without informed consent as the study satisfied the three criteria. First, it did not include pediatric patients aged <18 years. Second, it is realistically impossible to obtain consent from the research subjects during the research process. Third, there is no reason to assume that research participants will refuse consent, and the risk to research participants is extremely low, even if consent is waived. The study protocol satisfied these three criteria. The IRB approved the study protocol without obtaining informed consent.
We have added these comments in the Informed Consent Statement section.
- Line 76: Daugirdas equation – please, show this equation in this section.
Answer: Thank you for your comments. We have added the equation. Kt/Vurea was calculated using the Daugirdas equation, and the formula is as follows: Kt/Vurea = –Ln(R – 0.008*t) + (4 – 3.5*R)*UF/W (Ln, natural logarithm; R, postdialysis BUN/predialysis BUN; t, time; UF, ultrafiltration volume; W, postdialysis body weight).
- Line 86, 87 and 89: intensity of statin – please, change the word intensity into dose or dosage.
Answer: Thank you for your comments. We have revised “intensity of statin” to “dosage of statin” throughout the manuscript.
- Line 89: Medications such as aspirin, renin-angiotensin system blocker, and clopidogrel were also evaluated. If one or more prescriptions were identified a year before the evaluation of the HD quality assessment program, this was defined as the use of medication. Please, explain why you have chosen these groups of medications. This should be written in this section. Please, explain, CCI index in the text. All indexes should be explained in the text, they should be clear for the readed to follow the text.
Answer: Thank you for your comments. These medications are commonly used in patients on HD, especially those using statins, and can be considered confounding factors. Antiplatelet agents, such as aspirin or clopidogrel, can be used to treat various cardiovascular diseases such as myocardial infarction, angina, and peripheral vascular disease. Although these comorbidities were evaluated using ICD-10 codes in our study, diagnoses using ICD-10 codes can be associated with inaccuracies in diagnosis, and the use of these medications for primary prevention may also influence patient outcomes. Additionally, the use of RASBs is associated with favorable outcomes in patients taking the drug for secondary prevention of cardiovascular diseases, and the use of RASBs for primary prevention can also be associated with favorable outcomes despite incomplete conclusions regarding the association between the use of RASBs and patient mortality. Therefore, we used these factors as covariates to reduce their confounding effects and limitations associated with the inaccuracy of diagnosis.
We have added these comments in the Methods section.
Additionally, we have added relevant information for the comorbidities and scoring of Charlson Comorbidity Index. Detailed explanation is presented as an answer for your other comment.
- Line 94: Comorbidities were defined using the codes described by Quan et al. Please, explain it for the reader. Such information should be written, reades cannot look for such methodology in references. Methodology should be written clearly and in your manuscript it is not. Also, the inclusion and exclusion criteria should be written more clearly.
Answer: Thank you for your comments. In our study, the Charlson Comorbidity Index (CCI) included 17 comorbidities; MI, congestive heart failure (CHF), peripheral vascular disease, cerebrovascular disease, dementia, chronic pulmonary disease, rheumatologic disease, peptic ulcer disease, mild liver disease, diabetes mellitus (DM) without chronic complications, hemiplegia, renal disease, DM with chronic complications, any malignancy, moderate-to-severe liver disease, metastatic tumor, and acquired immune deficiency syndrome. The presence of comorbidities was defined using the ICD-10 codes listed in Table S2. All patients were considered to have renal disease due to HD (two points for moderate or severe renal disease). Finally, the CCI score was calculated using the 17 comorbidities mentioned above.
Table S2. The ICD–10 codes were used in Charlson Comorbidity Index
Comorbidities |
Codes |
Score |
Myocardial infarction |
I21, I22, I252 |
1 |
Congestive heart failure |
I43, I50, I099, I110, I130, I132, I255, I420, I425-I429, P290 |
1 |
Peripheral vascular disease |
I70, I71, I731, I738, I739, I771, I790, I792, K551, K558, K559, Z958, Z959 |
1 |
Cerebrovascular disease |
G45, G46, I60-69, H340 |
1 |
Dementia |
F00-03, G30, F051, G311 |
1 |
Chronic pulmonary disease |
J40-47, J60-67, I278-279, J701, J703, J684 |
1 |
Rheumatologic disease |
M05–06, M32–34, M315, M351, M353, M360 |
1 |
Peptic ulcer disease |
K25–28 |
1 |
Mild liver disease |
B18, K73, 74, K700-703, K709, K713-715, K717, K760, K762-764, K768–769, Z944 |
1 |
DM without complication |
E100-101, E106, E108-111, E116, E118-121, E126, E128-131, E136, E138-141, E146, E148-149 |
1 |
DM with complication |
E102-105, E107, E112-115, E117, E122-125, E127, E132-135, E137, E142-145, E147 |
2 |
Hemiplegia or paraplegia |
G81-82, G041, G114, G800, G830-834, G839 |
2 |
Any malignancy |
C00-26, C30-C34, C37-41, C43, C45-58, C60-6, C81-88, C90-97 |
2 |
Moderate to severe liver disease |
I850, I859, I864, I982, K704, K711, K721, K729, K765-767 |
3 |
Metastatic tumor |
C77-80 |
6 |
AIDS/HIV |
B20-22, B24. |
6 |
Abbreviations: ICD–10, International Classification of Diseases, 10th revision, Clinical Modification; DM, diabetes mellitus; AIDS/HIV, acquired immunodeficiency syndrome/human immunodeficiency virus
Additionally, we have added a study flowchart to more clearly display the inclusion and exclusion criteria of participants. Figure 1 is as follows:
Figure 1. Study flowchart
Results
Results section is well written and well organised. However,
- Line 139: what is PD abbrevation? Maybe I only cannot find it in the text.
Answer: Thank you for your comments. We have revised “PD” to “peritoneal dialysis (PD)” in the relevant line of Methods section.
- Please, do not write: patients with a prescription of statin, change it into e.g. patients taking statins. Prescribed treatment does not mean that patient take drugs.
Answer: Thank you for your comments, and we fully agree with the reviewer’s feedback.
Our study included only information about prescription status, rather than actual medication intake, to distinguish patient groups. Therefore, patients were categorized based on whether they had been prescribed medications and the number of days for which these prescriptions were issued. While an analysis based on actual medication intake could provide greater accuracy, our study exclusively utilized claims data related to prescription status, prescription quantities, prescription durations, and associated claims data, without including data on patients’ actual consumption or chart reviews. This approach could introduce a potential discrepancy between prescribed medications and their actual consumption, which is considered a significant limitation. Despite these limitations, it is worth noting that recent population-based studies have increasingly utilized claims data to classify and analyze patients based on their prescriptions. These studies serve as preliminary investigations and are conducted before randomized trials or similar studies are performed.
We have added these comments in the Discussion section.
Discussion
Discussion section is well written, but also it lacks more information on statins in general, as in introduction section. The manuscript will be much better with such information.
Answer: Thank you for your comments.
With the emergence of studies examining the association between cholesterol and cardiovascular diseases as well as mortality, efforts have been made to efficiently reduce cholesterol levels, especially low-density lipoprotein (LDL) cholesterol. The discovery of HMG-CoA reductase, a rate-controlling enzyme in cholesterol synthesis, has paved the way for these efforts. Researchers in Japan, including Akira Endo, successfully isolated compactin, a competitive inhibitor of HMG-CoA reductase, from fungi, and its effectiveness was confirmed when it was first applied in humans [1]. Subsequently, lovastatin became the first statin to be marketed, and over time, a total of 6 statins, including 2 semi-synthetic statins (simvastatin and pravastatin) and 4 synthetic statins (fluvastatin, rosuvastatin, pitavastatin, and atorvastatin), were developed and are currently in use [2]. Previous guidelines have emphasized that hypercholesterolemia is a well-known cardiovascular risk factor, and the use of statins for primary prevention is recommended based on atherosclerotic cardiovascular disease risk assessment [3]. Furthermore, two guidelines from the United States and Europe noted a direct correlation between cholesterol levels and atherosclerotic cardiovascular diseases in the general population, recommending statin therapy for secondary prevention [4]. Statins are commonly used as first-line therapies for lipid disorders, particularly in patients with elevated LDL-cholesterol.
We have added these comments in the Discussion section.
Added references
[1] Endo, A.A. historical perspective on the discovery of statins. Proc Jpn Acad Ser B Phys Biol Sci. 2010, 86, 484-493.
[2] Grundy, S.M.; Stone, N.J.; Bailey, A.L.; Beam, C.; Birtcher, K.K.; Blumenthal, R.S.; Braun, L.T., et al. 2018 AHA/ACC/AACVPR/AAPA/ABC/ACPM/ADA/AGS/APhA/ASPC/NLA/PCNA Guideline on the Management of Blood Cholesterol: Executive Summary: A Report of the American College of Cardiology/American Heart Association Task Force on Clinical Practice Guidelines. Circulation. 2019, 139, e1046-e1081.
[3] Arnett, D.K.; Blumenthal, R.S.; Albert, M.A.; Buroker, A.B.; Goldberger, Z.D.; Hahn, E.J., et al. 2019 ACC/AHA Guideline on the Primary Prevention of Cardiovascular Disease: Executive Summary: A Report of the American College of Cardiology/American Heart Association Task Force on Clinical Practice Guidelines. Circulation. 2019, 140, e563-e595.
[4] Virani, S.S.; Smith, S.C.; Jr, Stone, N.J.; Grundy, S.M. Secondary Prevention for Atherosclerotic Cardiovascular Disease: Comparing Recent US and European Guidelines on Dyslipidemia. Circulation. 2020, 141, 1121-1123.
Please, improve English in the manuscript, avoid word repetition in the text.
Answer: Thank you for your comment. English proofreading was carried out by two professional English proofreaders.

Round 2
Reviewer 2 Report
Authors properly replied to the comments. Minor English editing required.
Minor English editing recquired.